# Dissipation Kinetics and Safety Evaluation of Flonicamid in Four Various Types of Crops

**DOI:** 10.3390/molecules27238615

**Published:** 2022-12-06

**Authors:** Tao Zhang, Yue Xu, Xuan Zhou, Xiaojie Liang, Yang Bai, Fengshou Sun, Wenwen Zhang, Ning Wang, Xiuyu Pang, Yuekun Li

**Affiliations:** 1National Wolfberry Engineering Research Center, Wolfberry Science Research Institute, Ningxia Academy of Agriculture and Forestry Sciences, Yinchuan 750002, China; 2Shandong Provincial Key Laboratory for Biology of Vegetable Diseases and Insect Pests, College of Plant Protection, Shandong Agricultural University, Tai’an 271018, China; 3Research Center of Pesticide Environmental Toxicology, Shandong Agricultural University, Tai’an 271018, China; 4Department of Nutrition and Food Hygiene, School of Public Health, Shandong First Medical University & Shandong Academy of Medical Sciences, Tai’an 271016, China

**Keywords:** flonicamid, QuEChERS, dissipation, final residues, dietary risk assessment

## Abstract

The chemical insecticide flonicamid is widely used to control aphids on crops. Differences among crops make the universality of detection methods a particularly important consideration. The aim of this study was to establish a universal, sensitive, accurate and efficient method for the determination of flonicamid residues in peach, cucumber, cabbage and cotton. QuEChERS pretreatment was combined with ultrahigh-performance liquid chromatography-tandem mass spectrometry (UHPLC-MS/MS). A satisfactory recovery rate of 84.3–99.3% was achieved at three spiking levels, and the relative standard deviation (RSD) was 0.41–5.95%. The limit of quantification (LOQ) of flonicamid in the four matrices was 0.01 mg/kg. The residue and dissipation kinetics of flonicamid in four types of crops in various locations were determined by using the optimized method. The results showed that flonicamid had a high dissipation rate in the four different types of crops and a half-life in the different matrices and locations of 2.28–9.74 days. The terminal residue of flonicamid was lower than the maximum residue limit (MRL). The risk quotient (RQ) of flonicamid was 4.4%, which is significantly lower than 100%. This result shows that the dietary risk presented by using flonicamid at the maximum recommended dose is low and acceptable. The comprehensive long-term dietary risk assessment of flonicamid performed in this study provides a reference for the protection of consumer health and safe insecticide use.

## 1. Introduction

Cucumber (*Cucumis sativus* L.) belongs to the Cucurbitaceae family, and cabbage (*Brassica oleracea* L.) belongs to the Cruciferae family. Both of these vegetables are major products [1,2] that are widely distributed throughout China. Peach (*Prunus persica* L.) belongs to the Rosaceae family and accounts for a high proportion of fruit production in China [3]. Cotton (*Gossypium* spp.), belonging to the Malvaceae family, is one of the most important crops in the world economically and plays an important role in China’s economic development [4]. However, infestation by aphids significantly affects crop yield and quality, which severely limits the healthy development of related industries. Aphids have the characteristics of rapid reproduction and a short development cycle and have been treated with frequent pesticide application, resulting in severe aphid resistance to pesticides and difficulty in controlling these pests. Pyrethroid and neonicotinoid insecticides are commonly used to control aphids, but are also harmful to beneficial insects. Flonicamid is safe for humans, has low toxicity to fish and natural enemies, and exhibits no cross-resistance with pyrethroid and neonicotinoid insecticides. However, flonicamid poses certain risks to honey bees, so it should be used carefully when applied or sprayed in vegetable protection areas pollinated by honey bees [5]. Therefore, the use of flonicamid has become a major component of the control of crop aphids.

Flonicamid is a novel pyridine amide insecticide that exerts its effect by hindering pest sucking. Pests stop sucking soon after ingesting the pesticide and finally die of starvation [6]. Flonicamid has a high efficiency, low toxicity, strong internal absorption and long duration. The unique action mechanism of flonicamid enables the control of aphids and other piercing-sucking pests in vegetables, fruit, tea, grain and oil crops [7,8,9].

Pesticide use ensures crop yield and quality, but is associated with negative health risks [10,11]. More attention should be given to the problem of pesticide residues. To ensure safety and protect the environment, it is necessary to assess the residual behavior, field dynamic dissipation behavior and dietary risk of flonicamid in various crops. However, no studies have been performed to simultaneously assess the detection methods, dynamic dissipation behavior and dietary risk of flonicamid in vegetables, fruit, grain and oil crops. There are a large number of reports on analytical methods for flonicamid detection. Flonicamid has been determined in hops by acetonitrile extraction, C_18_ and solid-phase-extraction column purification, and liquid-liquid extraction [12]. The average recovery for flonicamid extraction from cucumber and apple using phosphate buffer and ethyl acetate can reach 82.9–104.1% [13]. Addition of sodium citrate and disodium citrate to a salting-out agent resulted in an average recovery of fipronamide from pepper of 78.4–109.3% [14]. The quantity of flonicamid in pepper was detected by UHPLC-Orbitrap-MS with an average recovery of 84–98% [15]. Flonicamid in cucumber was detected by acetonitrile extraction and C_18_ purification using acetonitrile and water as the mobile phase [16]. However, the abovementioned methods are complex and time-consuming. A simpler and more universal detection method was developed in this study: sample pretreatment and detection methods were optimized, resulting in the efficient and accurate detection of flonicamid in four different crops. The residue analysis and chronic risk assessment of flonicamid in apple, cabbage, honeysuckle and paprika have been reported in the literature [9,15,17,18]. However, as far as we know, there is no article discussing the dissipation rules on peaches, cucumbers, cabbage and cotton at the same time, which is of great significance for assessing their potential dietary intake hazards to consumers and ensuring food safety.

In this study, UHPLC-MS/MS with optimized QuEChERS sample pretreatment technology was used to simultaneously determine the concentrations of flonicamid in four different crops. This method is generally applicable to crops with high contents of sugar, water, pigment and oil. The established method was used to spray a 20% flonicamid suspension (SC) on four matrices in various regions, and the terminal residue and dissipation dynamics of flonicamid in the four matrices were evaluated. Finally, field experiments were performed, and the results were used to assess the long-term intake risk assessment of flonicamid to consumers. This study can provide data support for the establishment of guidelines for the safe use of flonicamid, as well as maximum residue limit standards. This data can also provide support and assistance for protecting consumer health and evaluating the quality and safety of agricultural production.

## 2. Materials and Methods

### 2.1. Chemicals and Reagents

A 96% flonicamid solution was provided by Hebei Xingbai Agricultural Technology Co., Ltd. (Hebei, China). Acetonitrile (UHPLC grade) and methyl alcohol (UHPLC grade) were purchased from Merck (KGaA, Darmstadt, Germany). Formic acid (UHPLC grade) was purchased from Shanghai Aladdin Biochemical Technology Co., Ltd. (Shanghai, China). Sodium chloride (NaCl) (analytical grade) was purchased from Shanghai Guangnuo Chemical Technology Co., Ltd. (Shanghai, China). Anhydrous magnesium sulfate (MgSO_4_) was purchased from Tianjin Kemeiou Chemical Reagent Co., Ltd. (Tianjin, China). PSA and C_18_ were purchased from Tianjin Agela Technologies Co., Ltd. (Tianjin, China).

### 2.2. Sample Extraction and Purification

All samples were pretreated using the QuEChERS method [19]. Then, 20 mL of acetonitrile were added to a homogenate of cucumber, cabbage, peach and cottonseed, and the resulting mixture was shaken for 5 min (the cottonseed samples were soaked in 10 mL of deionized water for 30 min in advance). NaCl and MgSO_4_ were added to the solution, and the mixture was shaken for 3 min. The mixture was centrifuged at 4000 r/min for 2 min, and 1.5 mL of the supernatant were prepared for purification.

Various quantities of PSA and MgSO_4_ were placed in a 2 mL centrifuge tube (30 mg of C_18_ were added for the cottonseed samples). The previously withdrawn supernatant (1.5 mL) was placed in a 2 mL centrifuge tube. The sample was shaken for 5 min and then centrifuged at 10,000 r/min for 2 min. The supernatant was diluted five times. The sample was passed through a 0.22 μm membrane filter. The membrane pore size was sufficiently small to filter out impurities other than the target compound and thereby reduce the influence of other matrices on the results. The weights of the agents used for homogenization, salting-out and purification by various treatments are shown in Appendix A.

### 2.3. UHPLC-MS/MS Parameters

UHPLC-MS/MS analysis of flonicamid was performed using a Waters ACQUITY UHPLC^®^BEH C_18_ column (2.1 mm × 50 mm; i.d., 1.7 μm) equipped with an electrospray ionization (ESI) source. The flow rate was 0.2 mL/min, and the injection volume was 1 μL. Formic acid (0.05%) in deionized water and methanol were used for gradient elution. The gradient elution parameters are listed in Appendix A. Reliable and fast separation of flonicamid was achieved under the optimized UHPLC conditions. Multiple reaction monitoring (MRM) was utilized to quantify and selectively detect the pesticide in a variety of crops. MS/MS was carried out using an electrospray ionization source in negative ion mode.

### 2.4. Preparation of Stock Solutions and Generation of Standard Curves

The 96% flonicamid solution was diluted with acetonitrile to prepare 1000 mg/L standard stock solutions, which were stored at −20 °C until use.

The sample matrices were treated with solvent standard solutions and matrix standard solutions at concentrations of 5, 1, 0.5, 0.1, 0.05, 0.01, 0.005 and 0.001 mg/L; the resulting mixtures were passed through a 0.22-μm membrane filter in preparation for UHPLC-MS/MS detection.

### 2.5. The Addition and Reclamation Test

Flonicamid standard solutions of three concentrations were added to blank samples with five replicates per level. The addition concentrations of flonicamid in the matrix were 0.01, 1 and 5 mg/kg; the addition recovery and relative standard deviation were determined.

### 2.6. Method Validation

A guidance document for the validation of analytical methods and quality control procedures for pesticide residues in food and feed was referred to in this study [20]. To verify the feasibility of the proposed method, the linearity, matrix effect (ME), accuracy, precision and limit of quantification (LOQ) of the method were determined. To evaluate the accuracy and precision of the method, flonicamid standard solutions at three concentrations were added to blank samples of the four crops. Five replicates were used for each treatment. The addition concentrations were 0.01, 1 and 5 mg/kg (the addition concentrations for the cottonseed blank sample were 0.01, 1 and 3 mg/kg), and the recovery rate and relative standard deviation (RSD) were calculated to determine the best combination of purifying agents. The sensitivity of the method was evaluated in terms of the LOD and LOQ. The LOQ of the proposed method was defined as the lowest spiked concentration quantification.

As co-extracts formed during the pretreatment process affect the accuracy of the results, it is necessary to calculate and exclude the matrix effect. The ME was calculated from the slopes of the solvent calibration curve and matrix-matched calibration curve [21]. The calculation formula is:ME/% = [(k_matrix_ − k_solvent_)/k_solvent_] × 100
where is the slope of the solvent calibration curve and is the slope of the matrix-matched calibration curve. If |ME| is less than 10%, there is no discernible matrix effect. If |ME| is greater than 10%, there is a clear matrix strengthening or weakening effect.

### 2.7. Field Trials

Field experiments were performed consisting of residual dissipation experiments and terminal residue experiments. Terminal residue experiments were carried out on flonicamid in peaches in Inner Mongolia, Shanxi, Beijing, Shandong, Henan, Jiangsu, Hunan and Guizhou. Residual dissipation experiments were carried out on peaches in Shanxi, Beijing, Shandong and Henan. Appendix A is a list of the sites, crop varieties, test types and planting methods of the four crops used in the experiments on residual dissipation and terminal residues. Aphids are the most registered control targets of flonicamid. Therefore, we selected common aphid damage targets and registered crops for the residue experiment.

The terminal residue experiment was performed on one treatment plot and one control plot. The area of each plot was not less than 50 m^2^. The treatment and control plots were set up in adjacent areas with isolation areas between the plots. The residual dissipation and terminal residue experiments were performed on the same plots. Four regions on which the residue dissipation test experiments were performed were selected to carry out terminal residue experiments.

The recommended maximum dose was used as the application dose. The entire plant was sprayed once before the occurrence of young larvae of aphids or at the initial peak stage. Samples were collected at fixed intervals after the last application. Three samples were collected for each treatment. During the first and last sample collections, samples from the control plot were collected at the same time for dynamic analysis of the residual dissipation experiment. Samples were collected two days after application for use in the terminal residue experiment. All the samples were maintained at −20 °C for testing. The application concentration of flonicamid for various crops was the recommended maximum dose, and samples for the residual dissipation and terminal residue experiments were collected on days set based on a safety interval.

### 2.8. Dietary Risk Assessment

The national estimated daily intake (NEDI, mg/kg/day) and risk quotient (RQ) for the long-term intake risk were calculated using the following formulas [22,23]:NEDI = STMRi × Fi
RQ (%) = NEDI/ADI × 100%

STMRi refers to the median concentration (mg/L) of the pesticide determined in a standard residue experiment on a food. Fi (kg/day) represents the average daily intake of special farm food for the general population in China, which is considered in conjunction with pesticide registration in China and the per capita dietary structure of Chinese residents. The recommended daily intake (0.192 mg/kg/day) is the acceptable daily intake of pesticide residues in China [24]. ADI is the acceptable daily intake of pesticide residues.

The higher the RQ is, the higher the pesticide residue is. An RQ > 100% indicates that the evaluated food presents an unacceptable health risk to consumers [25,26,27].

## 3. Results and Discussion

### 3.1. Optimization of UHPLC-MS/MS Conditions

It is very important to screen ionization conditions and chromatographic separation according to target analytes to improve the sensitivity and accuracy of analysis [28]. A point-spray positive ionization source (ESI+) mode was used for ionization scanning of flonicamid. Scanning was performed in multiple reaction detection mode. The parameters for the precursor ions and daughter ions were manually adjusted. Two characteristic ions with significant and stable responses were selected as sub ions. The collision energy of the characteristic ions was optimized. The determined mass spectrometry parameters are shown in Table 1.

An optimal mobile phase can considerably improve the peak shape in the chromatogram and the retention behavior of compounds in liquid chromatography systems. In this study, the gradient elution method was used to determine the elution efficiency of a 0.05% formic acid in the aqueous phase. The gradient elution procedure produces an enhanced peak shape in the chromatogram to improve the analysis efficiency. The largest flonicamid response was obtained using a mobile phase of 0.05% formic acid in the aqueous phase. This result showed that a low concentration of formic acid promoted the ionization of flonicamid. When 0.05% formic acid in water was used as the mobile phase system, the separation efficiency of flonicamid was high, and an ideal detection effect was achieved. A stable and strong signal response was detected within 6 min, with no discernible endogenous interference peaks. Therefore, this method has the advantages of high efficiency and speed.

### 3.2. Optimization of Sample Pretreatment

Acetonitrile is a general extraction solvent that is widely used in residue detection and has a high recovery rate for most pesticides [29]. Therefore, acetonitrile was used as the extraction reagent to maximize the extraction of residual pesticides in the matrix and minimize interferences. QuEChERS is commonly used as a pretreatment method because of simple, rapid operation and low cost [30,31,32]. In the QuEChERS sample pretreatment method, the extracted pesticide must be further purified to remove interferences from other extracts to increase the signal-to-noise ratio. (S/N) [33]. However, different combinations of purification materials need to be used for various crop types to optimize purification. As peach has a high sugar content, cucumber has a high pigment content, cucumber and cabbage have high water contents, and cottonseed has a high oil content, it is necessary to develop a general sample pretreatment method. PSA and C_18_ are the most commonly used solid adsorbents. PSA mainly removes polar impurities, such as organic acids, metal ions, fatty acids and sugars. C_18_ mainly removes nonpolar matrices, such as fats and sterols [34,35,36]. An optimal purification combination was selected according to the characteristics of the various purification materials used and test crops. Various combinations and quantities of purification materials were screened for various crops (vegetables: cucumber and cabbage; fruit: peach; grain and oil group: cotton). As cotton is rich in nonpolar matrices, such as sterols, the target compounds were purified by C_18_. The purification efficiency was evaluated in terms of the recovery rate. The highest recovery obtained was 70–120%. A 0.22-μm organic membrane filter was selected.

The recovery rates obtained by various purification combinations are shown in Figure 1. The recovery ranges obtained using various purification combinations (150 mg PSA + 50 mg MgSO_4_, 100 mg PSA + 100 mg MgSO_4_, and 50 mg PSA + 150 mg MgSO_4_) were 62.7–99.3%. Figure 1 shows that the quantities of PSA and MgSO_4_ used considerably impacted the recovery of flonicamid in the four matrices. The recoveries of 50 mg PSA and 150 mg MgSO_4_ in peach, cucumber, cabbage and cottonseed were 90%, 98.8%, 99.3% and 91.2%, respectively. These recovery rates were significantly higher than those obtained using other purification combinations. The results showed that the optimal purification efficiency was obtained using a combination of 50 mg PSA and 150 mg MgSO_4_. In conclusion, the optimal purifier combination for flonicamid in the peach, cucumber and cabbage samples was 50 mg PSA + 150 mg MgSO_4_. The optimal purifier combination for flonicamid in the cottonseed samples was 50 mg PSA + 150 mg MgSO_4_ + 30 mg C_18_.

### 3.3. Method Validation

A simple and convenient UHPLC-MS/MS method was established for the determination of flonicamid residues in vegetables, fruits, grain and oil crops. The validation parameters of the analytical method included the linearity, matrix effect, limit of quantitation, precision and accuracy. The linearity was evaluated in terms of the correlation coefficient (R^2^) between the calibration curve of flonicamid and the matrix-matched calibration curve. The linear range is within the acceptable standard for R^2^ ≥ 0.995 [37]. Table 2 shows the standard curve obtained using 6 data points. The R^2^ values for the standard curves of flonicamid solutions in peach, cucumber, cabbage and cotton were 1.000, 1.000, 0.999 and 0.999, respectively. The R^2^ of the standard curve of the flonicamid solution and the matrix-matched standard curve were all greater than 0.995 for the four crops, indicating an excellent linear relationship. The matrix effect (ME) refers to the influence of matrices other than the target compound on the response value of the target [38]. The MEs of flonicamid in peach, cucumber, cabbage and cotton were 1.07, 0.98, 0.98 and 0.98, respectively. The MEs of flonicamid in the four crops ranged from 0.98 to 1.07. There was no discernible matrix effect. Therefore, other matrices had a negligible influence on the response value of the target compound. The limit of quantitation of flonicamid in the four crops was 0.01 mg/kg. The accuracy and precision of the method were verified using the recovery test and relative standard deviation (RSD). Table 3 and Figure 2 shows the recovery rates of flonicamid in peach, cucumber, cabbage and cottonseed obtained using the optimal purifier combination. The recovery ranges of flonicamid in cucumber, cabbage, peach and cottonseed for the three spiking levels were 84.3–90.2%, 96.7–98.8%, 98.2–99.3% and 85.0–91.2%, respectively, and the corresponding RSD ranges were 1.6–2.6%, 0.41–5.95%, 1.07–4.12% and 1.6–4.6%. Thus, high recoveries of flonicamid at three spiking levels were obtained using the proposed method. The results show that the proposed method has good accuracy and precision. The evaluation of the linearity, matrix effect, limit of quantitation, precision and accuracy of the method shows that the proposed method can be used for the quantitative analysis of flonicamid in various crop samples. This method is suitable for the quantitative analysis of pesticide residues in vegetables, fruits, grain and oil crops. The method reduces the quantity of organic solvent used and improves environmental safety compared to conventional methods. The method has a high detection efficiency and a low cost. The method also has the advantages of high reliability, good stability and high sensitivity.

### 3.4. Dissipation and Terminal Residues of Flonicamid in Four Crops

The dissipation dynamics of flonicamid in four various crops in different regions were determined. Figure 3 shows curves for the dissipation dynamics, dissipation kinetics and flonicamid half-life. Figure 3 shows that the dissipation of flonicamid in the four crops follows first-order kinetics. The half-life ranges of flonicamid in peach, cucumber, cabbage and cotton are 3.44–7.63 days, 2.28–9.74 days, 3.30–8.82 days and 2.41–5.41 days, respectively. The half-life of flonicamid in the same crop is different for various regions. The volatilization and photolysis of pesticides caused by light and high temperature, scouring caused by rain and the physical and chemical properties of pesticides are all important factors affecting pesticide residues in plants [29]. The curve for the dissipation dynamics of flonicamid in cabbage is shown in Figure 3C. The half-life of flonicamid in cabbage in Guangdong is only 3.30 days, and the half-life of flonicamid in cabbage in Beijing is 8.82 days. The temperature, light and other environmental conditions in the two regions are quite different. The half-life of flonicamid in cabbage varies considerably across regions. The half-life and dissipation rate depend strongly on the regional location. The lower the latitude is, the shorter the half-life and the faster the dissipation rate are. The dissipation dynamics and half-life of flonicamid in other crops exhibits a similar trend. Cabbage is planted on land in summer. The temperature is high during the planting and growing period. After pesticide application, light, temperature and precipitation affect the dissipation of pesticides. These considerations also explain the experimental results for the dissipation dynamics and half-lives. The high dissipation rate of flonicamid in the four crops indicates a low likelihood of food safety problems due to pesticide residues in crops.

Table 4 shows the terminal residual quantity of flonicamid in the four crops in various regions. The experiment was performed using the recommended dosage and safety interval. At the first harvest, the ranges for the final residual quantity of flonicamid in peach, cucumber, cabbage and cotton were 0.01–0.13 mg/kg, 0.0158–0.134 mg/kg, 0.01–0.286 mg/kg and 0.01–0.246 mg/kg, respectively. At the second harvest, the ranges for the final residual quantity of flonicamid in peach, cucumber, cabbage and cotton were 0.01–0.022 mg/kg, 0.01–0.108 mg/kg, 0.01–0.129 mg/kg and 0.01–0.129 mg/kg, respectively. The final residues of flonicamid in the four crops in various regions were all within the maximum residue limit (MRL). Table 4 shows the terminal residues of flonicamid in the various crops in the tested areas arranged in ascending order. At the two harvests, the supervised trial median residue (STMR) of flonicamid in the four crops was at a low level. At the first harvest, the highest residue (HR) of flonicamid in peach, cucumber, cabbage and cotton was 0.130 mg/kg, 0.134 mg/kg, 0.286 mg/kg and 0.246 mg/kg, respectively. At the second harvest, the HR of flonicamid in peach, cucumber, cabbage and cotton was 0.022 mg/kg, 0.108 mg/kg, 0.129 mg/kg and 0.129 mg/kg, respectively. The HR was low for both harvests and considerably lower for the second harvest than for the first harvest, which indicated a low potential risk to consumers presented by flonicamid use in the four crops. The results showed that flonicamid can be used in vegetables, fruit, grain and oil crops according to the recommended dosage without leaving a residue exceeding the standard. These results were used for a dietary risk assessment of the four crops.

### 3.5. Dietary Risk Assessment

To evaluate whether using flonicamid as a pesticide is safe for consumers, we conducted a dietary risk assessment based on the residue data in food and the reference residue limit value (Table 5). The average weight of the general population in China is 63 kg [24]. The NEDI and RQ were calculated using the dietary intake and the average body weight data of the general population in China, following the principle of risk maximization combined with the residue of flonicamid in five foods. The dietary risk probability of flonicamid was assessed in terms of the RQ. The total NEDI of flonicamid was 0.20035 mg. This value is lower than the ADI determined by the European Union. The RQ was low at 4.4%. Flonicamid had an RQ of less than 100%, which is the acceptable level of flonicamid detected in registered crops, and is therefore safe for consumers. The results presented above show that the use of flonicamid at the recommended dose in cucumber, cabbage, peach and cotton does not present dietary risks to Chinese consumers.

## 4. Conclusions

A reliable, sensitive, general and accurate method based on QuEChERS and UHPLC-MS/MS was developed for the simultaneous determination of flonicamid in peach, cucumber, cabbage and cotton. Optimization and purification of the composite materials used in the method resulted in a high recovery, low RSDs and a low detection limit. The method was used to evaluate the dissipation and terminal residue of flonicamid in peach, cucumber, cabbage and cotton at several experimental sites. Flonicamid exhibited a high dissipation rate for various experimental sites and different matrices and a half-life between 2.28 and 9.74 days. The dissipation of flonicamid in various crops will certainly be influenced by environmental conditions. The terminal residue of flonicamid was less than the MRL. This result shows an acceptable exposure risk from applying flonicamid at the recommended dose. The low RQs of flonicamid indicate a low long-term dietary risk to Chinese consumers. A simple and rapid method is presented in this study for the simultaneous determination of flonicamid residues in four different crops. The data presented in this study provide guidance for the rational use of flonicamid and a reference for establishing the MRL of flonicamid in China.

## Figures and Tables

**Figure 1 molecules-27-08615-f001:**
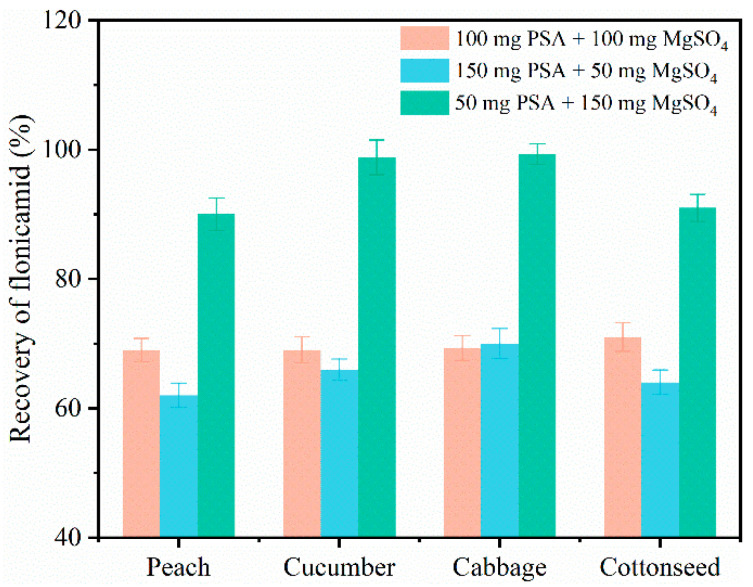
Effect of various purification combination materials on the recovery rate of flonicamid in four crops.

**Figure 2 molecules-27-08615-f002:**
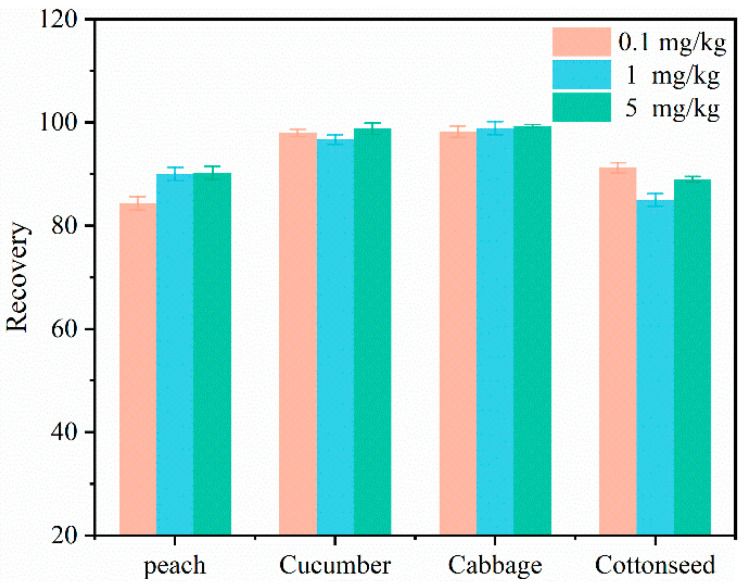
Recoveries of flonicamid in multiple matrices.

**Figure 3 molecules-27-08615-f003:**
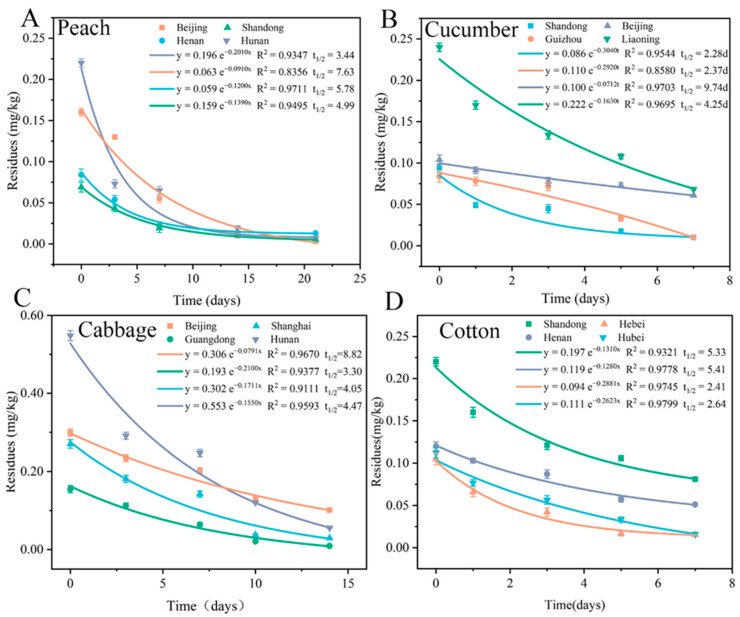
The dissipation dynamics, dissipation kinetics and half-life of flonicamid in (**A**) peach, (**B**) cu-cumber, (**C**) cabbage, (**D**) cotton.four crops.

**Table 1 molecules-27-08615-t001:** Collection parameters of the MRM of flonicamid.

Pesticide	Retention Time (min)	Precursor Ion (m/z)	Daughter Ion (m/z)	RF (V)	CE (V)
Flonicamid	1.8	230.1	203.0 174.1	116	16.37 17.17

**Table 2 molecules-27-08615-t002:** Linear regression equations, determination coefficients, matrix effects and LOQs of flonicamid.

Analyte	Matrix	*M* _e_	Linear Regression Equation	*R* ^2^	LOQ (mg/kg)
Flonicamid	Solvent	1.07	y = 2,057,469.1x − 2725.1	0.999	0.01
	Peach		y = 2,198,889.5x − 983.8	1.000	
	Solvent	0.98	y = 6,267,384.0x − 5559.5	1.000	0.01
	Cucumber		y = 6,159,848.6x − 397.5	1.000	
	Solvent	0.98	y = 11,194,433.4x − 26,234.8	0.999	0.01
	Cabbage		y = 10,975,030.1x − 17,906.0	0.999	
	Solvent	0.98	y = 9,414,632.4x − 8540.8	0.999	0.01
	Cottonseed		y = 9,262,984.6x − 7830.6	0.999	

**Table 3 molecules-27-08615-t003:** Recoveries and RSDs of flonicamid in different crops (n = 5).

Pesticide	Crop	Spiked Level/(mg/kg)					
		0.01		1		5 (3 *)	
		Recovery (%)	RSD (%)	Recovery (%)	RSD (%)	Recovery (%)	RSD (%)
Flonicamid	Peach	84.3	2.6	90.0	1.6	90.2	2.3
	Cucumber	98.0	5.95	96.7	0.41	98.8	4.31
	Cabbage	98.2	4.12	98.9	3.48	99.3	1.07
	Cottonseed	91.2	3.1	85.0	4.6	89.0	1.6

The band * is the spiked level of flonicamid in cottonseed.

**Table 4 molecules-27-08615-t004:** Terminal residues of flonicamid in four crops.

	Test Site	Application Dose	Application Times (Freq)	Harvest Interval (Days)	Residual Quantity (mg/kg)	STMR (mg/kg)	HR (mg/kg)
Peach	Inner Mongolia, Shanxi, Beijing, Shandong, Henan, Jiangsu, Hunan, Guizhou	37.5 (mg/kg)	1	14	<0.01, 0.01, 0.013, 0.015, 0.019, 0.049, 0.063, 0.13	0.017	0.130
				21	<0.01 (4), 0.013, 0.014, 0.015, 0.022	0.012	0.022
Cucumber	Liaoning, Inner Mongolia, Shanxi, Beijing, Shandong Tai’an, Henan, Shandong Qingdao, Anhui, Shanghai, Hunan, Hubei, Guizhou	75 (g/ha)	1	3	0.0158, 0.0192, 0.0202, 0.0218, 0.0427, 0.0447, 0.0520, 0.0571, 0.0717, 0.0764, 0.0777, 0.134	0.0484	0.134
				5	<0.01 (2), 0.0155, 0.0160, 0.0175, 0.0257, 0.0301, 0.0327, 0.0331, 0.0708, 0.0730, 0.108	0.0279	0.108
Cabbage	Shanxi, Beijing, Shandong, Henan, Anhui, Shanghai, Hunan, Jiangxi, Guangxi, Hubei, Guizhou, Guangdong	67.5 (g/ha)	1	7	<0.01 (2), 0.0121, 0.0155, 0.0170, 0.0879, 0.111, 0.112, 0.142, 0.202, 0.247, 0.286	0.0994	0.286
				10	<0.01 (5), 0.0135, 0.0263, 0.0291, 0.0556, 0.101, 0.121, 0.129	0.0199	0.129
Cotton	Hebei, Shandong, Henan, Anhui, Hunan, Jiangxi, Beijing, Hubei	60 (g/ha)	1	7	<0.01, 0.0119, 0.0175, 0.0889, 0.124, 0.142, 0.227, 0.246	0.0532	0.246
				10	<0.01 (2), 0.0135, 0.0151, 0.0163, 0.0572, 0.0930, 0.129	0.0157	0.129

**Table 5 molecules-27-08615-t005:** Chronic dietary risk assessment of flonicamid based on Chinese dietary patterns.

Food Classification	Fi(kg)	Reference Residue Limits	Sources	NEDI (mg)	ADI (mg)	Risk Quotient (%)
Rice and its products	0.2399	0.5	China	0.11995	ADI × 63	
Tubers	0.0495	0.2	China	0.00990		
Light vegetable	0.1837	0.0994	STMR	0.01826		
Fruits	0.0457	1	China	0.04570		
Vegetable oil	0.0327	0.2	China	0.00654		
Total	0.5515			0.20035	4.41	4.4

## Data Availability

Data is contained within the article and Appendix A.

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
