# Peer review of "Dissipation Kinetics and Safety Evaluation of Flonicamid in Four Various Types of Crops"

_molecules, 2022, doi:10.3390/molecules27238615_

Round 1

Reviewer 1 Report

Zhang et al. had established a universal, accurate and sensitive method for the determination of the residue of flonicamid in four crops. The feasibility of the method is fully discussed and verified by field experiments. Overall, this research was well designed and the authors have provided sufficient evidence to support the conclusions. However, a minor revision was required before publication. The main comments are listed as follows.

Comments

1In section 2.6, the addition concentration of flonicamid in cotton and peach was inconsistent with the results in Table 3. Please check it carefully.

2In section 3.4, the description was tedious. The dispersion of flonicamid in various experimental sites may be influenced by environmental conditions. The dissipation dynamics of pesticides will certainly be affected by the environment. Please reorganize the relevant sentences.

3In section 2.6, after ME appears, abbreviations will be used in the following text. Please correct other similar errors.

4In section 2.6, the area unit format is error, please check and modify.

5In section 3.3, the purification materials have been described in section 2.6, which does not need to be described again.

6In section 3.4, “The half-life of flonicamid in cabbage varies considerably 309 across regions. The half-life and dissipation rate depend strongly on the regional location across regions. The half-life and dissipation rate depend strongly on the regional location.” These two sentences are jumbled. Please combine them into one sentence.

Reviewer 2 Report

General Comments:

This manuscript represents the work of the authors to develop a general method for the extraction and quantification of the insecticide flonicamid in residues from  peach, cucumber, cabbage and cotton. Using QuChERS.  They also investigate the dispersion of this insecticide in various regions of the People's Republic of China and estimate the dietary risk of flonicamid to the general population from consumption of the above agricultural products.  Although the manuscript is generally well written and has scientific merit, some justifications in the introduction regarding the long term risk assessment are inadequate and some experimental details are lacking.

Specific Comments;

Introduction:

Line 51.  The authors state that flonicamid is safe for honey bees, however other research literature indicates otherwise.  The literature cited by the authors (reference 5) is inadequate and not appropriate.

Line 62.  The authors state that it is necessary to assess the residual and dissipation behavior concurrently with the dietary risk assessment.  They do not explain why this is important.  Each of these studies have been carried out independently in the scientific literature, why do they need to be presented in one manuscript?

Line 84.  Background references need to be included for the long term risk studies.

Materials and Methods:

Line 91.  Is the standard a certified standard?  What is the solvent of this solution?  How was the purity determined?

Line 101. Explain briefly how the samples were pretreated with QuEChERS.

Line 128.  What solvent was used and what specific matrix was used?

Line 130.  What type of membrane was used?  Nylon? PTFE?  

Line 148.  The description of LOQ seems to be the definition of the limit of detection (LOD).  Please define!

Lines 150-152.  What co-extracts form during the pretreatment process?  Define ME (matrix effect).  What solvent is used and what is used to make the matrix matched samples?

Line 173.  What is the recommended dose for the field studies?  

Line 192.  Define acceptable daily intake as ADI.

Results and Discussion:

Line 204.  The 2nd word “-6ions” means what?  Is this a typo?  Explain.

Line 206 table 1.  How can the daughter ions m/z be known to three decimal places when using a quadrupole instrument?  It should be the same as the precursor ion.  Please explain or correct this.

Line 231.  GCB is not defined nor is it used in any of the experiments described in this manuscript.  

Line 240.  Explain why a 0.22 membrane was selected?  What type?

Line 322.  Figure 3 is a bit too small and difficult to read.  Please enlarge.
